# Observations in the Spanish Mediterranean Waters: A Review and Update of Results of 30-Year Monitoring

Manuel Vargas-Yáñez [1,*], Francina Moya [1], Mariano Serra [2], Mélanie Juza [3], Gabriel Jordà [2], Enrique Ballesteros [1], Cristina Alonso [1], Josep Pascual [4], Jordi Salat [4], Vicenç Moltó [2], Elena Tel [5], Rosa Balbín [2], Rocío Santiago [2], Safo Piñeiro [2] and Mª Carmen García-Martínez [1]

[1] Instituto Español de Oceanografía, Consejo Superior de Investigaciones Científicas (IEO-CSIC), Centro Oceanográfico de Málaga, Puerto Pesquero de Fuengirola s/n, 29640 Fuengirola, Spain; francina.moya@ieo.csic.es (F.M.); enrique.ballesteros@ieo.csic.es (E.B.); cristina.alonso@ieo.csic.es (C.A.); mcarmen.garcia@ieo.csic.es (M.C.G.-M.)

[2] Instituto Español de Oceanografía, Consejo Superior de Investigaciones Científicas (IEO-CSIC), Centro Oceanográfico de Baleares, Muelle de Poniente s/n, 07015 Palma, Spain; mariano.serra@ieo.csic.es (M.S.); gabriel.jorda@ieo.csic.es (G.J.); vicenc.molto@ieo.csic.es (V.M.); rosa.balbin@ieo.csic.es (R.B.); rocio.santiago@ieo.csic.es (R.S.); safo.pineiro@ieo.csic.es (S.P.)

[3] Balearic Islands Coastal Observing and Forecasting System (SOCIB), Parc Bit Naorte, 07121 Palma, Spain; mjuza@socib.es

[4] Institut de Ciencies del Mar, Consejo Superior de Investigaciones Científicas, Passeig Marítim de la Barceloneta, 37–49, 08003 Barcelona, Spain; salat@icm.csic.es (J.S.)

[5] Instituto Español de Oceanografía, Servicios Centrales, C/Corazón de María, 8, 28002 Madrid, Spain; elena.tel@ieo.csic.es

\* Correspondence: manolo.vargas@ieo.csic.es

**Abstract:** The Instituto Español de Oceanografía (IEO, Spanish Institute of Oceanography) has maintained different monitoring programs in the Spanish Mediterranean waters (Western Mediterranean) since 1992. All these monitoring programs were unified in 2007 under the current program RADMED (series temporales de datos oceanográficos en el Mediterráneo), which is devoted to the in situ multidisciplinary sampling of the water column of coastal and open-sea waters by means of periodic oceanographic campaigns. These campaigns, together with a network of tide-gauges, are part of the IEO Observing system (IEOOS). In some cases, the temperature and salinity time series collected in the frame of these monitoring programs are now more than 30 years long, whereas sea level time series date to the beginning of the 1940s. This information has been complemented with international databases and has been analyzed in numerous works by the Grupo mediterráneo de Cambio Climático (GCC; Mediterranean Climate Change Group) for more than 20 years. These works have been devoted to the detection and quantification of the changes that climate change is producing on the physical, chemical, and biological properties of the Spanish Mediterranean waters. In this work, we review the results obtained by the GCC since 2005 in relation to the changes in the physical properties of the sea: water column temperature, salinity, and density, heat content, mixed layer depth, and sea level. Time series and results are updated from the last works, and the reliability of the existing time series for the detection of climatologies and long-term trends are analyzed. Furthermore, the different sources of uncertainty in the estimation of linear trends are considered in the present work. Besides this review and update of the results obtained from the data collected in the frame of the IEOOS, we conduct a review of the existing monitoring capabilities from other institutions in the Spanish Mediterranean waters and a review of results dealing with climate change in the Spanish Mediterranean obtained by such institutions. In particular, we include a review of the results obtained by SOCIB (Servicio de Observación y Predicción Costero de las Islas Baleares; Balearic Islands costal observing and forecasting system) in relation to the study of marine heat waves and the warming of the sea surface, and the results corresponding to the intense warming of the Catalan continental shelf at L'Estartit oceanographic station. All these results evidence that the surface Spanish Mediterranean waters are warming up at a rate higher than that affecting the global ocean (>2 °C/100 years). This warming and a salinity increase are also observed along the whole water column. Marine heat waves are increasing their intensity, frequency, and duration since

1982, and coastal sea level is increasing at a rate of 2.5 mm/yr. The salinity increase seems to have compensated for the warming, at least at surface and intermediate waters where no significant trends have been detected for the density. This could also be the reason for the lack of significant trends in the evolution of the mixed layer depth. All these results highlight the importance of monitoring the water column and the necessity of maintaining in situ sampling programs, which are essential for the study of changes that are occurring throughout the Spanish Mediterranean waters.

**Keywords:** climate change; Spanish Mediterranean waters; Western Mediterranean; ocean observing systems

## 1. Introduction

Climate change is one of the biggest threats to the Mediterranean Sea. Its water masses exhibit increased temperature and salinity over the entire depth range since mid-twentieth century [1–3]. Sea Surface Temperature (SST) has increased since the early 1980s at a rate three or four times higher in the Mediterranean Sea than in the global ocean [4], and the intensity, duration, and frequency of Marine Heat Waves (MHWs) are substantially increasing [5]. Sea level has risen since the end of the nineteenth century [6] with an acceleration in the sea level trends since the early 1990s [4,6–8].

Projections from Regional Climate Models show that the temperature, salinity, and sea level of the Mediterranean Sea will continue to increase during the twenty-first century, with variable trends depending on the emission scenario considered [6,9–12].

All these changes already detected, and the projections commented above, make the observation of the oceans, in general, and that of the Mediterranean waters in particular, a task of paramount importance. Oceanographic data are essential, on the one hand, for detecting and quantifying changes in the marine environment, and on the other, for the calibration and validation of numerical models as well as for assimilating into models to improve the predictability.

Monitoring of physical variables of surface waters has experienced a huge development, mainly with the generalized use of radiometers and altimeters since the late 1970s and early 1990s, respectively. Thanks to this, Essential Ocean Variables (as defined by the Global Ocean Observing System), such as SST and sea level, can be monitored with a global coverage and high spatial and temporal resolution (see [13] for a review of the evolution of remote sensing applied to oceanography). These remotely sensed data are suitable for analyzing ongoing changes to daily, monthly, inter-annual, and long-time scales and are assimilated into models with different spatio-temporal scales. However, they are limited to the surface or first tens of meters of the water column and are missing changes and processes of the greatest importance in the present context of climate change. In particular, it should be considered that most of the heat absorbed by the oceans because of global warming has been stored in the upper 700 m, and changes can be detected below 2000 m depth [14]. This is also the case with the Mediterranean Sea where intermediate and deep waters exhibit increased temperature and salinity. The calculation of the thermosteric and halosteric contributions to the sea level rise [15,16] or the depth of the mixed layer [17] are other examples of variables that require the knowledge of the vertical distribution of temperature and salinity along the water column. Many other examples of the need for the monitoring of the water column could be given considering the biogeochemical variables. For example, the extension of the minimum oxygen layer could be increasing [18], or the nutrient supply to the euphotic layer could be decreasing in some areas of the world ocean because of a decrease in the vertical mixing linked to an intensification of the water column stratification [5,19].

Focusing on the Mediterranean, the Spanish Mediterranean waters constitute a large geographical area within the Western Mediterranean (WMED). Their coastal length is 3457 km, and the surface covered by waters in the continental shelf and upper continental

slope is around 53,000 km$^2$ (waters above 500 m depth, including the Balearic Islands). Besides their large geographical extension, there are scenarios of several oceanographic processes of great importance for the understanding of the functioning of the Mediterranean Sea: The Alboran Sea is the first of the Mediterranean basins that receives the Atlantic waters and is a preferential place for detecting water mass changes imported from the nearby Atlantic Ocean. It has an intense mesoscale activity with strong thermohaline fronts such as the Alboran Front and the Almería-Oran Front [20–23]. This basin is also the last one crossed by the Mediterranean waters before flowing through the Strait of Gibraltar. The Balearic Channels are considered as a choke point for the exchange of water masses between the northern and the southern WMED [24–26], and the continental shelf in the Catalan Sea, the Balearic Islands, and even the surroundings of Cape Palos could be sites for western intermediate water formation [27–29].

Considering the relevance of this geographical area of the WMED, and the importance of the systematic and periodic monitoring of the water column, both in coastal and open-sea waters, this work intends to analyze the present state of the observing systems maintained by the Instituto Español de Oceanografía (IEO; Spanish Institute of Oceanography) and also by other institutions. Particular attention will be paid to the IEO Observing System (IEOOS) and its suitability for the estimation of seasonal climatologies and the detection of long-term changes associated with climate change. We also address the different sources of uncertainty in such calculations and review and update the results obtained along the past decades. To give a broader view of the changes that climate change is causing on physical variables in the Spanish Mediterranean, we complete this review with the results obtained by the Sistema de Observación y Predicción Costero de las Islas Baleares (SOCIB; Balearic Islands Observing and Forecasting System) and L'Estartit oceanographic station. Although the main focus is on the role of in situ observations, we also consider the results obtained from remotely sensed data that are relevant for understanding the changes that the Spanish Mediterranean waters are undergoing in the present context of climate change.

## 2. Materials and Methods

This section summarizes, on one hand, the main observation networks and some of the available databases of oceanographic data in the Mediterranean Sea (Section 2.1). On the other hand, Section 2.2 reviews some of the data analysis methods used for obtaining the main results presented in the results section. Section 2.1 includes the observing systems maintained by IEO (Sections 2.1.1–2.1.3) and those operated by SOCIB, ICM-CSIC (Institut de Ciencies del Mar; Consejo Superior de Insvetigaciones Científicas), and Puertos del Estado (Sections 2.1.4–2.1.7). Finally, it summarizes some of the databases used in the present work and in others and are also reviewed in this study. Section 2.2 describes the data analysis methods. It is divided into two sections, one devoted to the description of the construction of the time series that have been analyzed for the detection of long-term trends Section 2.2.1 and another Section 2.2.2 which describes the different sources of uncertainty present in the estimation of long-term changes.

### 2.1. Observation Networks and Datasets

2.1.1. RADMED Project

The IEO initiated the multidisciplinary monitoring of the Spanish Mediterranean waters in 1992 with the launch of the Ecomálaga project. This project included three transects of oceanographic stations distributed on the on–off shore direction in the waters around Málaga Bay (P, M, and V in Figure 1). These transects were visited on a three-monthly basis (seasonally). Ecomálaga was followed by Ecobaleares in 1994 (B in Figure 1). This project was made of a north–south transect at the southern continental shelf of Mallorca Island. This project had a monthly periodicity until 2000 and three-monthly afterwards. In 1996, two more projects joined the IEO observing system of the Mediterranean waters: Ecomurcia, in the area of Cape Palos, and CIRBAL (Circulación en los Canales Baleares; Circulation in the Balearic Channels) in the Balearic Channels (CP and C, respectively, in

Figure 1), also with a three-monthly periodicity and a multidisciplinary character. All these projects were unified in 2007 under the umbrella of the RADMED project (Series temporales de datos oceanográficos en el Mediterráneo; Time series of oceanographic data in the Mediterranean Sea). This latter project completed the spatial coverage of previous programs adding new transects and extending seawards the previous ones (transects S, CG, T, BNA, MH, and the Cabrera deep station in Figure 1). At present, RADMED is made of a set of 86 oceanographic stations that are distributed along on–off shore transects, with the only exception of those stations within the Balearic channels, which form two triangles (C in Figure 1). The RADMED stations cover the continental shelf and slope of the Spanish Mediterranean from Málaga to Barcelona (including the Balearic Islands) and also include some deep stations in the open sea. The sampling has a three-monthly periodicity and is multidisciplinary, including CTD (conductivity–temperature–depth) casts at all the stations, and nutrient, chlorophyll, phytoplankton, and zooplankton sampling at some selected stations at discrete depths [30].

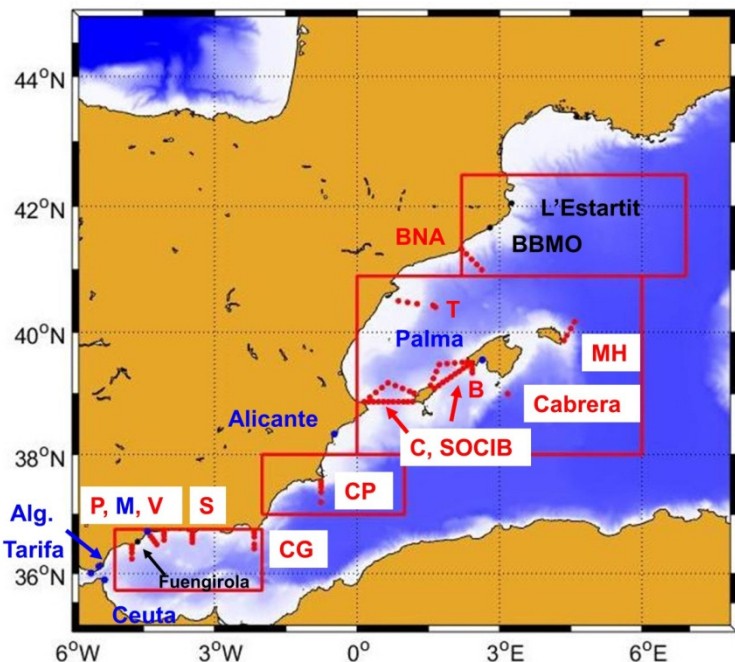

**Figure 1.** Red dots are the position of the RADMED project stations. Letters close to the transects stand for Cape Pino (P), Málaga (M), Vélez (V), Cape Sacratif (S), Cape Gata (CG), Cape Palos (CP), Balearic Channels (C), Mallorca (B), Tarragona (T), Barcelona (BNA), and Mahón (MH). Cabrera deep station is named as "Cabrera". Blue dots and blue labels indicate the position of coastal tide-gauges: Alegiras (Alg.), Tarifa, Ceuta, Málaga (M), Alicante, and Palma. Black dots show the positions of the Fuengirola temperature time series, L'Estartit oceanographic station and tide-gauge, and the Blanes Bay Microbiological Observatory (BBMO). The base of the triangles in the Balearic Channels also correspond to the glider lines and the seasonal surveys maintained by SOCIB.

### 2.1.2. Fuengirola Beach Temperature Time Series

Since 1984, daily measurements of sea surface temperature are carried out in Fuengirola Beach (Figure 1). Since December 2020, water samples at the same location are also collected for the measurement of chlorophyll-a concentration.

### 2.1.3. The IEO Tide-Gauge Network

The IEOOS [31] is completed in the Mediterranean waters with a network of tide-gauges at Tarifa, Algeciras, Ceuta, Málaga, and Palma (Figure 1). Sea level data from the Alicante tide-gauge, operated by the Instituto Geográfico Nacional and reconstructed by [32], have also been used. Monthly time series of sea level data from the Permanent Service for Mean Sea Level were also used for reconstructing the IEO sea level series

following the methodology described in [8]. Additionally, since 2020, a new experimental tide-gauge network (VENOM, Variabilidad espacial del nivel del mar en el Mediterráneo Occidental; spatial variability of sea level in the Western Mediterranean) has been deployed providing high-density coverage around the Balearic Islands (http://mareografo.ieo.es/, accessed on 1 April 2023).

### 2.1.4. L'Estartit Oceanographic Station

L'Estartit oceanographic station is located 4 km offshore L'Estartit, in the continental shelf of the Catalan Sea (Figure 1). Sea temperature is measured with a weekly or even higher periodicity (60 to 90 samples per year) at discrete depth levels from the sea surface to the bottom (80 m) since 1973. This is the longest uninterrupted time series of oceanographic data in the Mediterranean Sea [33]. However, in situ sampling, L'Estartit station is completed with a coastal tide-gauge and a meteorological station. This station is operated and maintained by the volunteer oceanographer and meteorologist, Josep Pascual, with partial support from the ICM-CSIC.

### 2.1.5. The Balearic Island Coastal Observing and Forecasting System

SOCIB is a multi-platform and integrated ocean observing and forecasting system that provides streams of oceanographic data, added value products, and forecasting services from the coastal areas to the open ocean [34,35]. Both physical and biogeochemical data are collected, mainly in the WMED, through multiple observational platforms equipped with multiple sensors. In the open ocean, the SOCIB observational network includes: (1) A fleet of nine autonomous underwater gliders equipped with a suite of sensors able to collect both physical and biogeochemical observations. These gliders have carried out a semi-continuous endurance line in the Balearic Channels since 2011 (coinciding with the base of the triangles in Figure 1) to monitor the water mass exchanges between the northern and southern sub-basins, their relation to the circulation variability, and their impact on ecosystem variations [4,25,34]; (2) A long-term monitoring program, with seasonal multidisciplinary transects along the Balearic Channels. These transects also coincide with the base of triangles in Figure 1 and are carried out using the R/V SOCIB; (3) Two oceanographic buoys located in the Ibiza Channel and the bay of Palma; (4) Two high-frequency coastal radar stations with antennas overlooking the Ibiza Channel [36,37]; (5) Lagrangian observational platforms through an annual deployment of three Argo profiling floats (within the framework of the Euro-Argo ERIC program) and eight surface drifters (as part of the Global Drifter Program) in the WMED as a contribution of the Global Climate Observing System and Global Ocean Observing System; (6) Animal-borne instruments with tracked sea turtles which provide information on essential biodiversity variables and contribute to knowledge-based marine conservation [38].

### 2.1.6. Puertos del Estado

The observational component of Puertos del Estado (Portus) system in the Mediterranean Sea consists of seven deep water buoys (able to measure currents, SST, salinity, wind, atmospheric pressure, air temperature, and waves), six coastal buoys (waves and SST), two high-frequency radar systems (Catalan coast and Strait of Gibraltar), and 16 tide-gauges. This comprehensive observing system is fully integrated into the modeling component of Portus and serves data for multiple socio-economic sectors with the main customer being the Spanish Port system. Real-time data access is provided via Portus website. Nevertheless, these data will not be analyzed in the present work and are mentioned in order to describe the present monitoring capabilities in the Spanish Mediterranean waters.

### 2.1.7. The Blanes Bay Microbiological Observatory

The Blanes Bay Microbiological Observatory is a station located 0.5 miles from Blanes coast (see Figure 1) over a bottom depth of 20 m. This station is visited on a monthly basis

by ICM-CSIC and is focused on the study of the microbiological component of the marine ecosystem.

### 2.1.8. Other Datasets

MEDAR/MEDATLAS (Mediterranean Archeology and Rescue/Mediterranean Hydrographic Atlas) database was the result of the European Union concerted action between 20 data centers in countries of the Mediterranean and Black Sea regions. This data base contains 36,054 CTD profiles, 88,453 bottle casts, and 161,848 bathythermograph profiles. These data are available at SeaDataNet (https://www.seadatanet.org/, accessed on 22 February 2022). Those temperature and salinity profiles corresponding to the four regions marked in Figure 1 by red rectangles have been used by the GCC [3,17] and in this work (see Section 3.1).

Mixed Layer Depth (MLD) values from Argo profilers were collected from mixedlayer.ucsd.edu/database [39], accessed on 22 February 2022. The Argo data are part of the Global Ocean Observing System (https://doi.org/10.17882/42182#56126, accessed on 22 February 2022).

Monthly gridded temperature and salinity data were obtained from the Met Office Hadley Centre observations datasets (version EN.4.2.1; [40]). This data set offers monthly data on a $1° \times 1°$ grid with 42 vertical levels. These data were used for calculating the steric contribution to sea level in grid points close to the locations of tide-gauges from 1940 to 2019.

In order to complete the information concerning the temperature of the surface layer, satellite daily SST data were collected since 1981 from the National Oceanographic and Atmospheric Administration (NOAA). The data set used is the "High-resolution Blended Analysis of Daily SST and Ice" (https://psl.noaa.gov/data/gridded/data.noaa.oisst.v2.highres.html, [41]) accessed on 18 October 2021, with a spatial resolution of $0.25° \times 0.25°$ in latitude and longitude. These data were used for the analysis of linear trends. We also used daily SST data from reprocessed (REP) and near real-time (NRT) high-resolution satellite products distributed by the Copernicus Marine Service. The REP (SST_MED_SST_L4_REP_OBSERVATIONS_010_021 1; https://doi.org/10.48670/moi-00173) and NRT (SST_MED_SST_L4_NRT_OBSERVATIONS_010_004_a_ V2 2; https://doi.org/10.48670/moi-00172; accessed on 31 March 2023) satellite products in the Mediterranean Sea provide optimally interpolated estimates of SST into a regular grid ($1/20° \times 1/20°$ and $1/16° \times 1/16°$, respectively) at daily resolution [42,43] since 1982.

The long-term sea level reconstruction based on tide-gauge data developed by [7] is also used to provide a more complete picture of coastal sea level trends in the WMED. This reconstruction is based on an optimal interpolation method in which the correlation between tide-gauge data and all coastal points was determined from the outputs of a numerical model. The reconstruction covers the period 1884 to 2019 with a monthly resolution and is available at https://doi.org/10.1594/PANGAEA.945345 accessed on 1 April 2023.

### 2.2. Data Analysis Methods

#### 2.2.1. Construction of Temperature and Salinity Time Series

Temperature, salinity, density, and ocean heat content time series from the RADMED stations have different starting dates. The oldest time series date to 1992, and the most recent ones to 2007. These time series were extended backward in time using the MEDAR/MEDATLAS database. We searched for all the available temperature and salinity profiles in the MEDAR/MEDATLAS database in small geographical areas close to each of the RADMED stations. The result was that the amount of available data was very low. Then, we divided our area of study into four large geographical areas represented in Figure 1 with red rectangles. The large extension of these four geographical areas is intended to allow the gathering of a large amount of data. However, we could not distinguish local or small-scale differences in the trends or long-term changes in the properties of water

masses. Yet, these squares could be expected to be representatives of four regions with distinctive properties of their water masses [3]: Alboran Sea, Cape Palos, Balearic Islands, and the northernmost region that will be called hereafter Northern Sector. The choice of these regions is a trade-off between the number of available data and the resolution of regional differences, and raised the question of the reliability of the trends that could be inferred from such time series.

Despite the large size of the selected regions, the number of available data was very low. Figure 2 shows the number of available salinity data at 0, 600, and 1200 m depth at the Alboran Sea and the Northern Sector, respectively. It is worth noting that there were frequent periods with no data for such large areas during several consecutive years. This problem worsens with increasing depth.

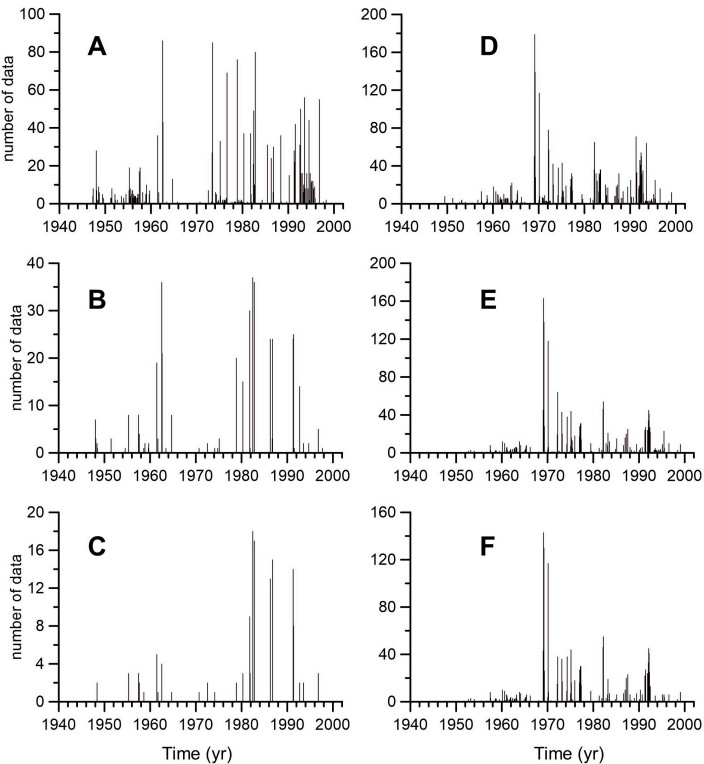

**Figure 2.** (**A–C**) show the number of data available for each month of the year at 0, 600, and 1200 m depth within the square corresponding to the Alboran Sea. (**D–F**) show the same results for the Northern Sector (see red rectangles in Figure 1).

The scarcity of data in the MEDAR/MEDATLAS database and the three-monthly sampling of RADMED project did not allow us to construct time series with a high time resolution (monthly time series, for instance). Therefore, we finally constructed three-monthly time series grouping and averaging all the temperature and salinity profiles corresponding to the same season and year within each of the four regions shown in Figure 1 (red rectangles) and for 23 discrete depth levels (0, 10, 20, 30, 50, 100, 150, 200, 300, 400, 500, 600, 700, 800, 900, 1000, 1200, 1400, 1500, 1750, 2000, 2250, and 2500 m). Previous to 1992, these averages were obtained only from MEDAR/MEDATLAS data. From 1992 to 2000, these averages included both MEDAR and RADMED data, and from 2000, they were made only of RADMED data.

The seasonal cycle increases the variance of time series, mainly in the upper 100 or 200 m of the water column, and should be removed before the estimation of long-term trends. This can be easily done following different procedures. We could average the four three-monthly data corresponding to each year and obtain annual time series for each depth level, or we could estimate a climatological seasonal cycle averaging all the data corresponding to the same season of the year within our time series. Then, we could

subtract the corresponding seasonal value from each data point of the time series obtaining time series of residuals or anomalies. Once again, some issues stemmed from data scarcity.

In order to explain the methodology, Figure 3A shows a detail of the time series of surface temperature at the Alboran Sea region corresponding to years 1982 and 1983. Black dots show the four seasonal values corresponding to these two years, and blue dots are the annual averages for both 1982 and 1983. Nevertheless, despite the large geographical areas considered for grouping the data, there were frequent gaps in these three-monthly time series. Even in the case of the RADMED project and the previous IEO monitoring programs, where the sampling was systematic and periodic, the final time series presented gaps because some campaigns or some of their parts could not be completed because of bad weather conditions, breakdowns of the oceanographic vessels, etc. In Figure 3B, it is considered a hypothetical case in which the summer value is missing in the year 1982, and the winter value is missing in 1983. There are several ways in which these gaps can be dealt with. In the first method, we could simply average the available data, ignoring the missing ones. This method is named hereafter as "ignorance". Red dots show the new annual mean values calculated when summer 1982 and winter 1983 data were missing. As expected, the new annual means (red dots) were biased with respect to the real mean values (blue dots). If the winter gaps were more frequent during the first (second) half of the series and the summer gaps were more frequent during the second (first) half, then the initial annual values would be positively (negatively) biased, and the second half of the time series would be negatively (positively) biased, reducing (enhancing) the estimated trends. On the other hand, if the season of the gaps is randomly distributed, the effect would be an increase in the variance of the time series as more extreme annual values would be artificially obtained (for instance, anomalous high annual means would be calculated if winter values were missing). This variance increase would make it difficult to detect statistically significant trends.

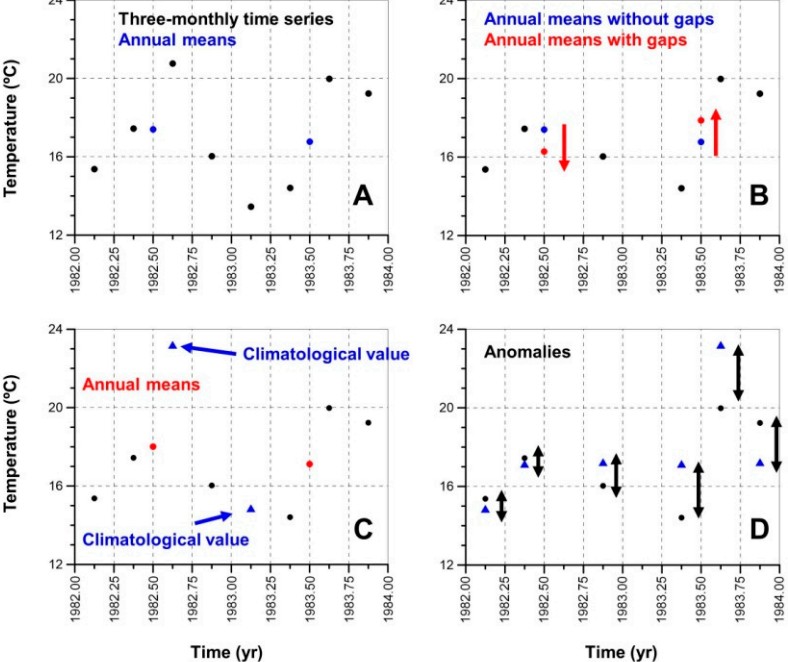

**Figure 3.** Black dots in (**A**) are the surface temperature values from a three-monthly sampling from satellite SST data in Málaga Bay for the years 1982 and 1983. Blue dots are the annual mean values for those years. (**B**) shows the same temperature values as in (**A**), eliminating the summer 1982 and the winter 1983 values. Blue dots are the annual mean values presented in (**A**), and red dots are the annual means calculated after the elimination of summer 1982 and winter 1983. In (**C**), the missing values have been substituted with the climatological averages calculated for the period 1982–2021

(blue triangles). Red dots are the new annual values after substitution using the climatologies. (**D**) illustrates the representative anomalies method. The climatological values corresponding to each season are plotted with blue triangles. Black arrows show the anomalies for each season of the year.

Figure 3C illustrates a second method. We could calculate the climatological seasonal cycle and fill the gaps with the corresponding climatological value. In this case, we assume that the missing value did not deviate from the climatology, and therefore, this method is known in the literature as "zero anomaly" [44].

Figure 3D exemplifies the third method. The anomaly or deviation from the climatology is calculated for the available data for each year. An annual anomaly is calculated averaging these anomalies, ignoring the missing data. This method implicitly assumes that the anomalies from the sampled seasons are representative of the anomaly of the complete year, and ignores that the anomaly corresponding to the missing season could have been different. Therefore, this method is known as "representative anomalies" [44].

Another possibility of processing the temperature and salinity time series could be considered as a modification of the "zero anomaly" method. Besides the calculation of the seasonal climatologies, a linear trend could be estimated for each depth level by means of a least square fit. The missing data would be replaced by the corresponding seasonal climatological value including the drift of such climatology associated with the linear trend (in the case that it does exist). This method will be named here as "zero anomaly + trend".

Refs. [45–47] have shown that XBT (Expendable Bathythermograph) data could induce some biases in the trend estimation. For this reason, we constructed annual time series of temperature and salinity for each region (Alboran, Cape Palos, Balearic Islands, and Northern Sector) and depth level, using the different methodologies described above, and including and excluding XBT data [48].

For each depth level, we had four annual time series corresponding to the four different methods described above. Finally, the time series obtained were averaged for three layers: surface (0–150 m), intermediate (150–600 m), and deep layer (600 m–bottom), averaging the temperature and salinity values within each depth range weighted by the thickness of the layers defined by the discrete depth levels [48]. Each annual vertical profile could also present gaps at discrete depth levels. Once again these spatial gaps were filled using zero or representative anomalies. In this way, we generated eight different time series for each layer corresponding to the four methods used for filling gaps in the time series, and the possibility of using zero or representative anomalies for filling the gaps in the vertical profiles. We also obtained time series of temperature, salinity, and density for each of the three layers (surface, intermediate, and deep) and averaged for the four geographical areas. We could have gaps for some of the layers at some of the four areas. These gaps were filled using the same method (zero anomaly or representative anomaly) that had been used for filling the gaps during the vertical average of the different layers. In this way, we also generated eight different time series for each layer that represent the average behavior of the complete area of study. In a similar way, heat content time series were also calculated for each layer and for the whole water column, integrating the temperature of the water column times the density and the specific heat.

Time series of MLD were calculated for each station using the threshold method. Two different approaches were followed: MLD was calculated using both a 0.3 °C temperature and a 0.03 kg m$^{-3}$ density thresholds as in [17].

Daily SST time series from satellite data were collected for all the grid points within each of the four studied areas (red squares in Figure 1). These time series were averaged for constructing monthly time series. Climatological seasonal cycles were calculated and then subtracted from the monthly time series for obtaining time series of anomalies.

### 2.2.2. Sources of Uncertainty

Linear trends were estimated fitting a straight line by means of least squares to each time series. Confidence intervals were calculated at the 95% confidence level. The



possible auto-correlation of the time series was taken into account substituting the degrees of freedom by the effective degrees of freedom, which were calculated using the auto-correlation function estimated for the time series [49,50]. In this way, when a time series was analyzed, the linear trend could be expressed as an interval [L, U], with L being the lower limit and U the upper one. However, for each variable, geographical area and layer, several time series were available as a result of the use of the different methodologies described above, and several trends and 95% confidence intervals were obtained for each of these time series. There is no clear way to determine which method is the right one in the sense that it provides the most accurate result. The final approach was to consider the minimum value of the lower limits (Lmin) and the maximum value of the upper limits (Umax). Finally, the estimated trend was considered as [Lmin, Umax]. In this way, the confidence intervals for the linear trends take into account both the uncertainty associated with the internal variability of the analyzed variable (variance of the time series) and the uncertainty linked to the selection of the data processing method.

The uncertainty in the trend estimation is also increased by the variable dates of the sampling. If the RADMED stations were always sampled the same day for each season of the year, then the different values recorded on different years would simply respond to the inter-annual variability of the sampled variable. On the contrary, different surveys corresponding to different years, but the same season of the year, were sampled on different dates. Figure 4 shows the distribution of dates for one of the stations from the Alboran Sea in the RADMED project. Averaging all the dates, it can be considered that the climatological seasonal cycle estimated from these time series corresponded to the average values for the distribution of dates. The winter climatological value corresponded to the month 2.3, the spring value to the month 5.3, the summer one to 7.6, and the autumn climatology to 11.0. Each data point in our time series is sampled on a different date that can differ from the average date. For instance, the climatological spring values corresponded to the beginning of May. If a campaign was carried out during mid-June, it will show a deviation with respect to the climatology, which will be caused by the natural inter-annual variability of the analyzed variable, but also because mid-June values could depart significantly from May values.

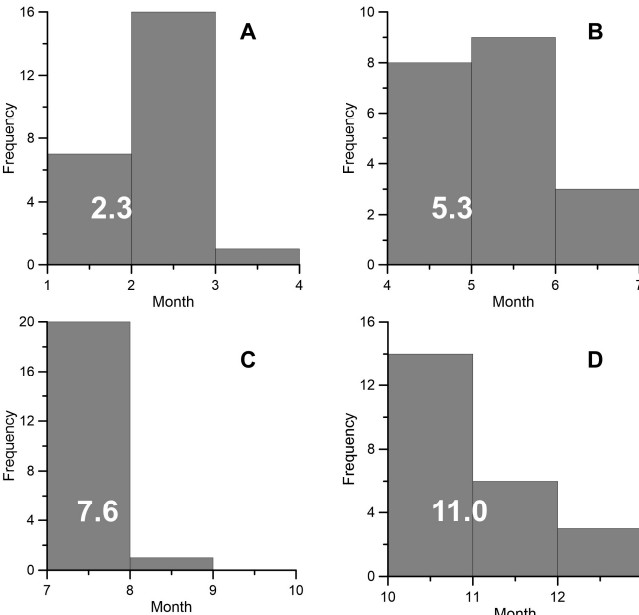

**Figure 4.** Frequency distribution of the dates corresponding to all the RADMED cruises in the Málaga Bay from 1992 for each season of the year: (**A**), winter, (**B**), spring, (**C**), summer, and (**D**), autumn. Inserts in each figure show the average date for each season.

All these issues cast some doubts on the reliability of the available time series for the estimation of long-term trends, and even climatological seasonal cycles. These questions were analyzed by means of a SST time series with daily temporal resolution. This time series was subsampled with a monthly time step (on the 15th day of each month) and three-monthly, with a fixed sampling day at the central date of each season: 15 February, 15 May, 15 August, and 15 November. Finally, it was also sampled on a three-monthly basis, but with a sampling day that was chosen randomly from the distribution of dates of the RADMED project and shown in Figure 4. This random sampling is similar to the real one in RADMED project as it selected a different day for each different season and year, and included the existence of gaps with the same frequency as in the original data set. We analyzed the seasonal cycles and trends obtained from the daily time series and from those that had been subsampled.

## 3. Results

### 3.1. Temperature, Salinity, Density, and Heat Content Trends

Figures 5 and 6 show the time series of potential temperature and salinity, respectively, averaged for the four geographical areas analyzed in this work. Left column in these figures shows those time series obtained with the different methodologies described above in Section 2.2.1 (see also [48,50]) for the surface (0–150 m, A), intermediate (150–600 m, C), and deep layers (600 m–bottom, E). These figures correspond to the time series constructed without the use of XBT data (see Section 2.2.1). Similar results were obtained for those series that included such data. The right columns in Figures 5 and 6 show the time series constructed as the mean values of the time series obtained from different methodologies (black lines). The gray-shaded areas show the range between the minimum and maximum values corresponding to different methodologies.

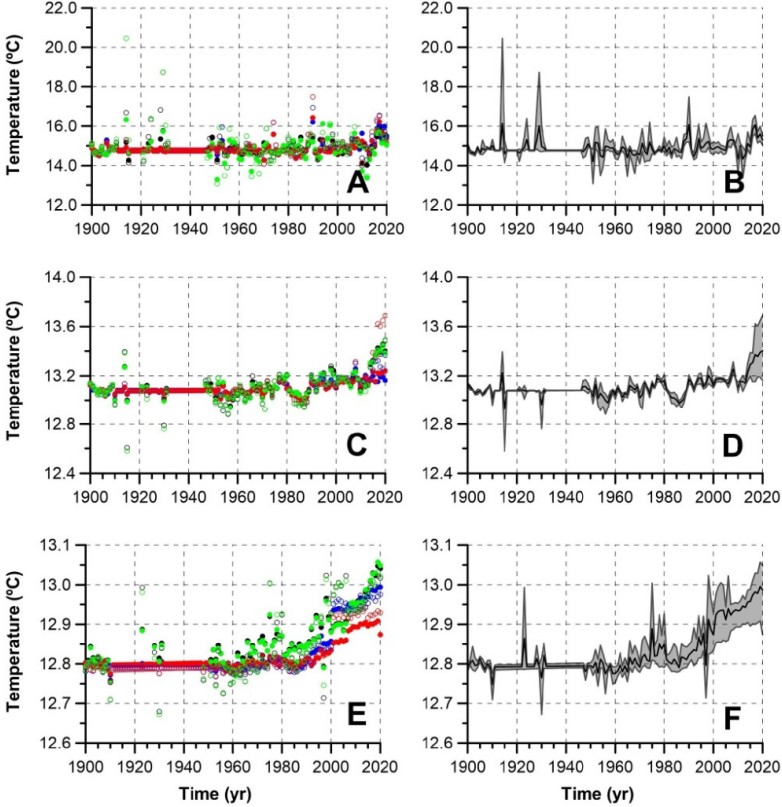

**Figure 5.** (**A**) shows all the potential temperature time series corresponding to different data processing methods for the surface layer (0–150 m) averaged for the four geographical areas analyzed in this work:

Alboran, Cape Palos, Balearic Islands, and Northern Sector. Black, blue, red, and green symbols correspond to representative anomalies, zero anomalies, zero anomalies + trend, and ignorance, respectively. Filled symbols correspond to the use of zero anomaly for the vertical and horizontal average, and open symbols to representative anomalies. Black line in (**B**) is the mean value of all the time series shown in (**A**). The gray-shaded area shows the range of values corresponding to the different methodologies. (**C**,**D**) show similar results, but for the intermediate layer (150–600 m), and (**E**,**F**) are the same, but for the deep layer (600 m–bottom).

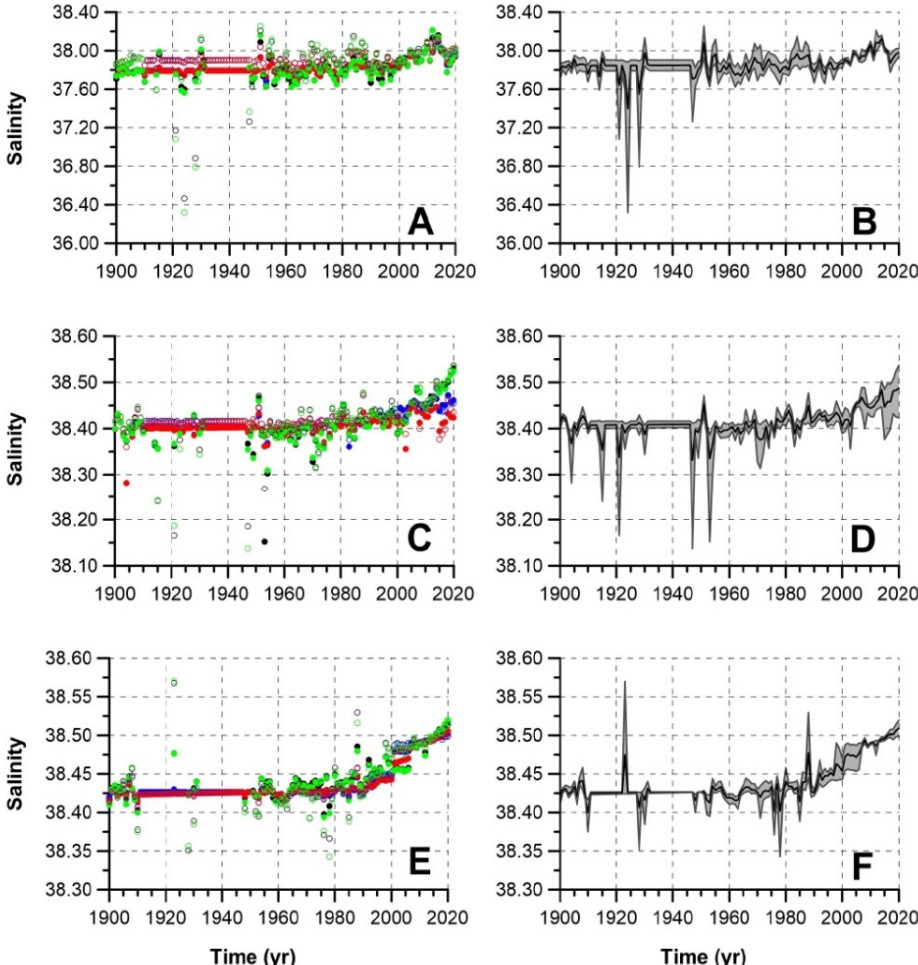

**Figure 6.** (**A**) shows all salinity time series corresponding to different data processing methods for the surface layer (0–150 m) averaged for the four geographical areas analyzed in this work: Alboran, Cape Palos, Balearic Islands, and Northern Sector. Black, blue, red, and green symbols correspond to representative anomalies, zero anomalies, zero anomalies + trend, and ignorance, respectively. Filled symbols correspond to the use of zero anomaly for the vertical and horizontal average, and open symbols to representative anomalies. Black line in (**B**) is the mean value of all the time series shown in (**A**). The gray-shaded area shows the range of values corresponding to the different methodologies. (**C**,**D**) show similar results, but for the intermediate layer (150–600 m), and (**E**,**F**) are the same, but for the deep layer (600 m–bottom).

Figure 7 shows the heat content averaged for the four geographical areas for the surface (7A), intermediate (7B), and deep layer (7C), and for the whole water column (7D). Different colored dots show the different methodologies used for the data processing (see Section 2.2.1). These figures do not include XBT data, although, as in the previous case, calculations including XBT data provided similar results.

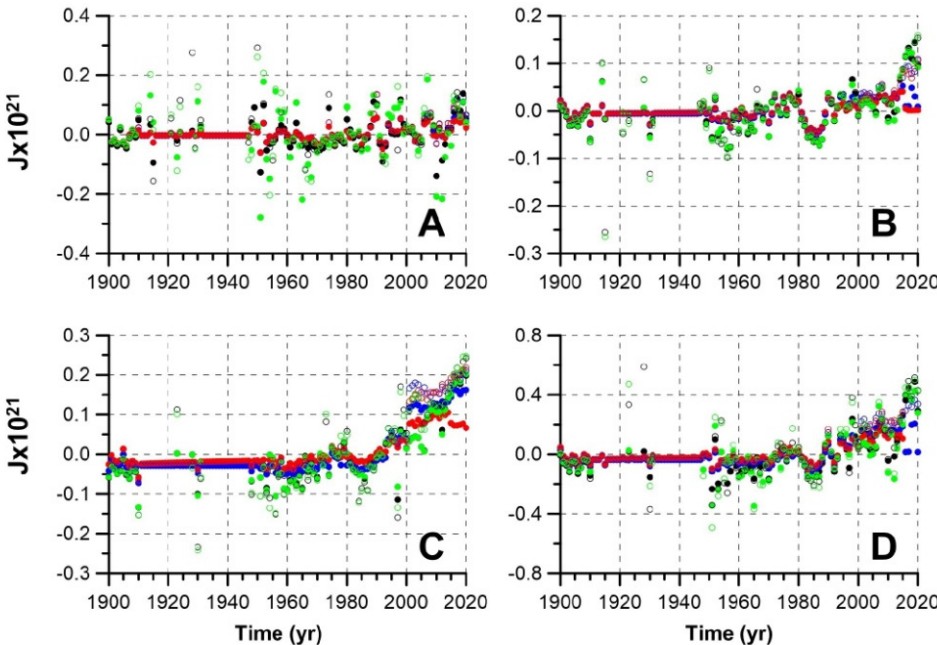

**Figure 7.** (**A**) shows the heat content time series corresponding to the different data processing methods for the surface layer (0–150 m) averaged for the four geographical areas analyzed in this work: Alboran, Cape Palos, Balearic Islands, and Northern Sector. (**B**,**C**) show similar results, but for the intermediate (150–600 m) and deep layers (600 m–bottom). (**D**) shows the heat content time series for the whole water column. Heat content is expressed in $J \times 10^{21}$.

95% confidence intervals [L, U] were calculated for the linear trends of each time series. Finally, we obtained a confidence interval for each variable and layer as the minimum of the lower limits and the maximum of the upper ones ([Lmin, Umax], see Section 2.2.2). These confidence intervals were estimated using all the data processing methods and both the time series including and excluding XBT data. These results are presented in Table 1 for temperature, Table 2 for salinity, Table 3 for density, and Table 4 for heat content, averaging the four geographical areas. The different areas (Alboran, Cape Palos, Balearic Islands, and Northern Sector) exhibited a similar behavior. Nevertheless, detailed trends for each region are presented in the supplementary material.

**Table 1.** Linear trends for the potential temperature of the surface, intermediate, and deep layers and for the whole water column averaged for the four geographical areas of Alboran, Cape Palos, Balearic Islands, and Northern Sector. Columns 3 and 5 show the 95% confidence intervals considering both the uncertainty associated with the natural variability of the system and that associated with the choice of the data processing method, i.e., [Lmin, Umax] (see Section 2.2.2). Columns 2 and 4 show the mid-points of the 95% confidence intervals. Trends are expressed in °C/100 years.

| Potential Temperature Trends (°C/100 Years) | | | | |
|---|---|---|---|---|
| | **1900–2020** | | **1945–2020** | |
| 0–150 | 0.34 | [−0.09, 0.76] | 0.50 | [−0.22, 1.22] |
| 150–600 | 0.08 | [0.00, 0.15] | 0.23 | [0.10, 0.36] |
| 600–bottom | 0.13 | [0.06, 0.19] | 0.25 | [0.16, 0.35] |
| Water column | 0.33 | [0.08, 0.58] | 0.12 | [−0.14, 0.39] |

**Table 2.** Same as in Table 1, except for salinity. Trends are expressed in $(100 \text{ years})^{-1}$.

| Salinity Trends $(100 \text{ Years})^{-1}$ | | | | |
|---|---|---|---|---|
| | 1900–2020 | | 1945–2020 | |
| 0–150 | 0.11 | [0.06, 0.16] | 0.23 | [0.08, 0.39] |
| 150–600 | 0.03 | [0.01, 0.05] | 0.09 | [0.04, 0.14] |
| 600–bottom | 0.05 | [0.04, 0.07] | 0.11 | [0.09, 0.13] |
| Water column | 0.08 | [0.04, 0.12] | 0.19 | [0.10, 0.29] |

**Table 3.** Same as in Table 1, except for density. Trends are expressed in $\text{kg m}^{-3}/100$ years.

| Potential Density Trends $(\text{kg m}^{-3}/100 \text{ Years})$ | | | | |
|---|---|---|---|---|
| | 1900–2020 | | 1945–2020 | |
| 0–150 | −0.01 | [−0.13, 0.11] | 0.03 | [−0.17, 0.24] |
| 150–600 | 0.01 | [−0.02, 0.03] | −0.01 | [−0.08, 0.06] |
| 600–bottom | 0.01 | [0.01, 0.02] | 0.03 | [0.02, 0.04] |
| Water column | −0.08 | [−0.19, 0.02] | −0.02 | [−0.13, 0.09] |

**Table 4.** Same as in Table 1, except for heat content. Trends are expressed in $\text{W/m}^2$.

| Absorbed Heat $(\text{W/m}^2)$ | | | | |
|---|---|---|---|---|
| | 1900–2020 | | 1945–2020 | |
| 0–150 | 0.03 | [−0.02, 0.08] | 0.05 | [−0.08, 0.17] |
| 150–600 | 0.03 | [0.00, 0.07] | 0.11 | [0.03, 0.18] |
| 600–bottom | 0.15 | [0.07, 0.22] | 0.34 | [0.22, 0.45] |
| Water column | 0.21 | [0.05, 0.37] | 0.46 | [0.20, 0.72] |

Trends have been calculated for the complete period (1900–2020) and for the period 1945–2020 when the amount of data increased considerably. Data were very scarce during the first half of the twentieth century, and time series were biased towards the climatological values. Nevertheless, Figures 5–7 show stable temperature, salinity, and heat content values until 1960 when positive trends seem to appear clearly until the end of the series, with the only exception of a brief cold period during the beginning of the 1980s decade. When linear trends were calculated for the 1945–2020 period, salinity showed positive and significant trends for all the layers. Temperature trends were positive and significant for this period in the intermediate and deep layers. The mid-point of the confidence interval was positive for the temperature trends of the surface layer. Nevertheless, this trend was not statistically different from zero. Density showed a significant trend for the deep layer, whereas these trends were not significant at the surface and intermediate layers. The behavior of the heat content reflects that of the temperature time series. Although all the layers absorbed heat, the trends were not statistically significant for the surface layer. The whole water column absorbed heat at a rate of between 0.20 and 0.72 $\text{W/m}^2$ from 1945. This range decreased to [0.05, 0.37] $\text{W/m}^2$ when it was calculated for the period 1900–2020 because of the temperature stability during the first half of the twentieth century.

*3.2. Surface Temperature Trends from Satellite Measurements, L'Estartit Oceanographic Station, and Fuengirola Time Series*

Figure 8B shows the climatological seasonal cycles for the satellite SST, for each of the four studied areas. Figure 8C–F present the corresponding time series of anomalies for the period 1981–2021. Trends represented by the slope of straight lines fitted by means of least squares are included (see inserts in Figure 8C–F). The uncertainty is expressed by means of

95% confidence intervals, taking into account the auto-correlation of the time series. In all the cases, the SST had a strong and significant trend. Such trends increased northwards from a minimum value of 2.4 ± 0.5 °C/100 yr in the Alboran Sea to a maximum value of 3.9 ± 0.6 °C/100 yr in the Northern Sector.

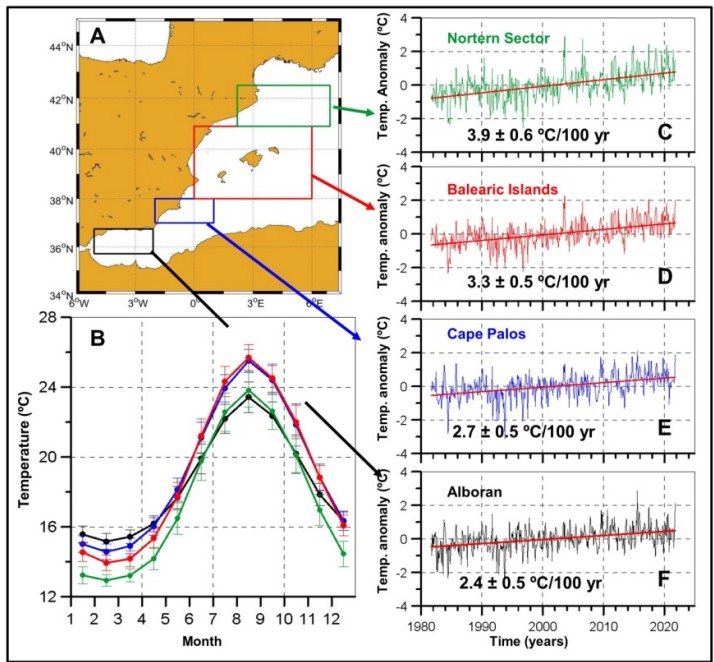

**Figure 8.** (**A**) shows the position of the four areas of study. Seasonal cycles for the SST of the four areas are presented in (**B**). The color of the curves coincides with the color used for the rectangles in (**A**). (**C**–**F**) are the time series of temperature anomalies for the four regions.

The intense SST trends, evidenced in Figure 8, were also observed in the whole water column of L'Estartit continental shelf. Figure 9A–D show the temperature anomalies for 0, 20, 50, and 80 m depth, respectively, in L'Estartit oceanographic station. Linear trends are inserted in these figures. Note that the temperature trend at the sea surface is 2.8 ± 0.4 °C/100 years, which, taking into account the uncertainty of these estimations, is close to the value obtained from satellite measurements for the Northern Sector. This trend at L'Estartit sea surface did not change significantly when it was calculated for the period 1981–2021, yielding a value of 2.7 ± 0.6 °C/100 years.

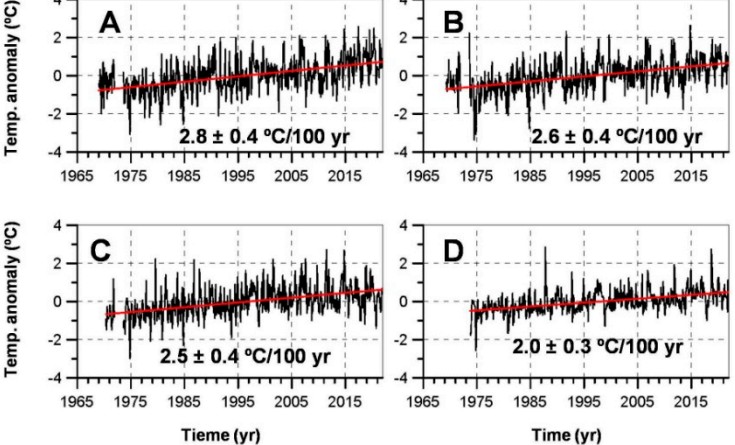

**Figure 9.** Time series of temperature anomalies at L'Estartit oceanographic station at 0 (**A**), 20 (**B**), 50 (**C**), and 80 m (**D**) depth (black lines). Linear trends expressed in °C/100 year have been inserted

in these plots. The uncertainty corresponds to the 95% confidence interval taking into account the auto-correlation of the time series. The red lines are the straight lines fitted by means of least squares for estimating the linear trends.

The temperature at the sea surface of Fuengirola Beach also showed an intense warming trend from 1985 to 2022 (Figure 10). In this case, the trend was $1.78 \pm 0.01$ °C/100 yr. This trend is lower than those estimated for L'Estartit waters, confirming the south–north gradient evidenced by satellite results.

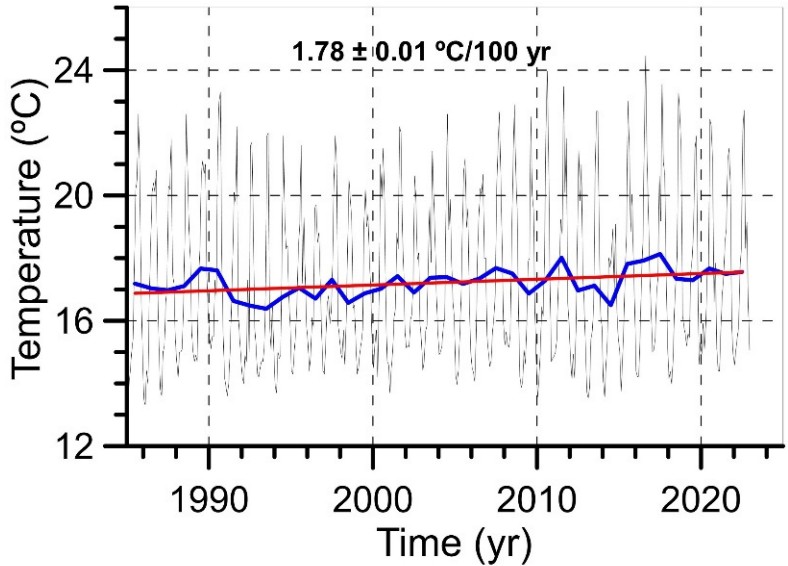

**Figure 10.** Monthly (black line) and annual (blue line) sea surface temperature time series from Fuengieola Becah. Red line shows the linear trend. The uncertainty is expressed by the 95% confidence interval taking into account the auto-correlation of the time series.

All these trends in the surface waters of the Spanish Mediterranean and in the shallow continental shelf of L'Estartit station simply reflect the intense warming that the Mediterranean Sea is experiencing both in the western and eastern sub-basins, with SST trends of $3.6 \pm 0.2$ °C/100 yr and $4.6 \pm 0.2$ °C/100 yr, respectively [4].

*3.3. Marine Heat Waves*

Marine heat waves (MHWs) in the Mediterranean Sea have been studied by [5]. These extreme events are considered to occur when SST exceeds a threshold (the local 90th percentile of long-term climatology) during at least five consecutive days [51]. Ref. [5] used the period 1982–2015 for calculating the threshold, and found that during the period 1982–2020, the mean and maximum intensity of MHWs, their duration, and intensity increased in all the sub-basins of the Mediterranean with linear trends that ranged, depending on the sub-basin considered, between 0.06 and 0.13 °C/decade (mean intensity), 1.23–3.82 °C/decade (maximum intensity), 1.2–3.8 days/decade (mean duration), 1.1–1.8 events/decade (frequency of MHWs), and 14–32 days/decade (total days).

In 2022, unprecedented MHW events were observed, particularly in the western Mediterranean Sea [52]. In the present paper, we report updated trends of MHW characteristics over the period 1982–2022, including the two recent years 2021 and 2022 to the time series from [5]. Trends of MHW characteristics reached maximum values of 0.18 °C/decade (mean intensity), 0.65 °C/decade (maximum intensity), 12.35 days/decade (mean duration), 2.4 events/decade (frequency), and 42 days/decade (total days), respectively. Figure 11A,B show the change for the annual total days and mean intensity of MHWs over the recently updated period 1982–2022 over the whole basin. While the trends in MHW total days are the highest in the eastern Mediterranean Sea (in particular, Levantine basin, northern Aegean Sea, middle-eastern Ionian Sea, and southern Adriatic Sea), trends

in MHW intensity reached the highest value in the north-western Mediterranean, northern Adriatic Sea, and northern Aegean Sea.

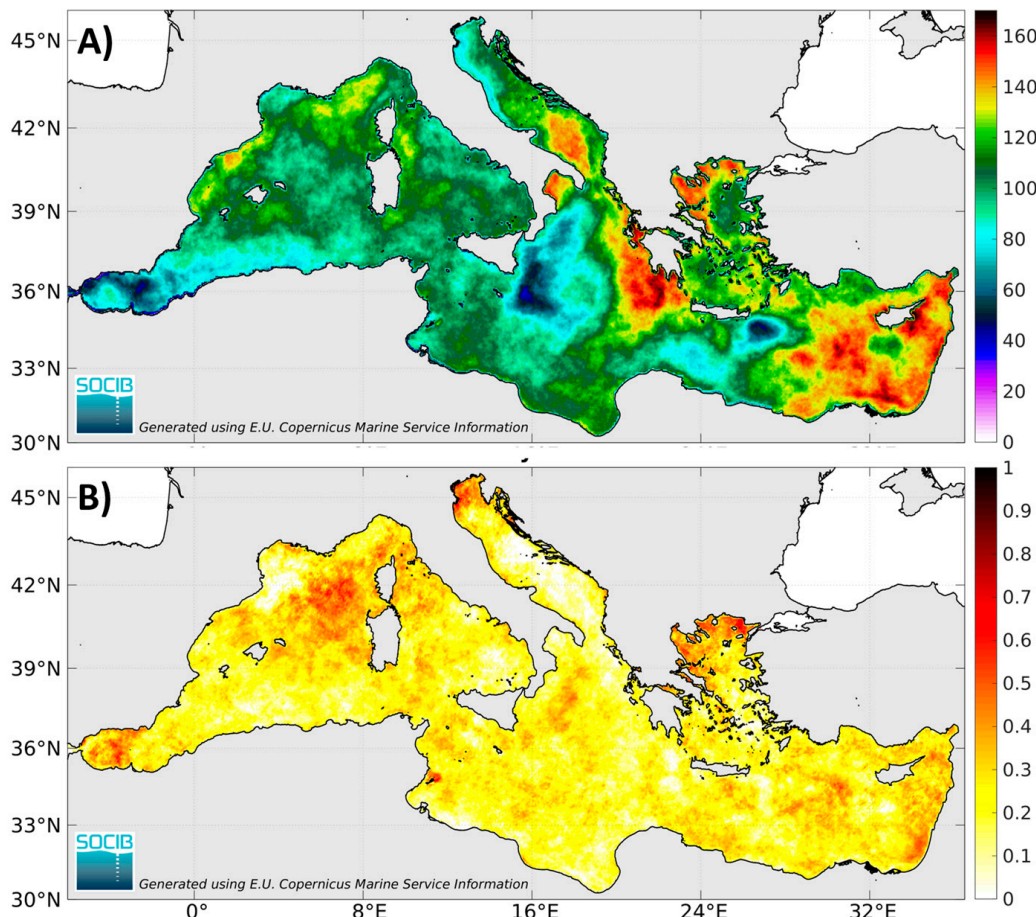

**Figure 11.** Change over the period 1982–2022 of MHWs annual total days (in days, (**A**)) and mean intensity (in °C, (**B**)) in the Mediterranean Sea. Source: https://apps.socib.es/subregmed-marine-heatwaves/ [52], accessed on 1 June 2023.

Besides the surface signature of MHWs, recent studies have shown their depth propagation into the ocean interior and their impact on properties of the water column [5,53].

### 3.4. Mixed Layer Depth Trends

Ref. [17] analyzed time series of MLD for each of the oceanographic stations from the RADMED project. This analysis was repeated for MLD values calculated using both the temperature and density threshold methods. In most of the cases the trends were not statistically significant, and positive and negative trends alternated. MLD values from Argo profilers were collected from mixedlayer.ucsd.edu/database [39]. These data were grouped for the four analyzed areas. Table 5 shows the trends estimated from the temperature and density threshold methods in [17].

The trends for the MLD were not statistically significant in the Alboran Sea, Cape Palos, and Northern Sector, despite the method used for its calculation. In the Balearic Islands, there was observed a significant shallowing of the MLD.

**Table 5.** Linear trends and 95% confidence intervals for the evolution of the MLD in the four selected geographical areas: Alboran Sea, Cape Palos, Balearic Islands, and Northern Sector. Trends are expressed in m/yr. Positive values indicate deepening of the mixed layer, and negative values indicate shallowing of the mixed layer. Columns 2 and 3 show the initial and final years of the time series analyzed. Columns 4 and 5 are the trends corresponding to MLD time series calculated by means of the temperature and density thresholds, respectively. This table is taken from [17].

|  | Initial Year | Final Year | t-Threshold | d-Threshold |
|---|---|---|---|---|
| Alboran Sea | 2006 | 2021 | $0.1 \pm 0.9$ | $0.1 \pm 0.8$ |
| Cape Palos | 2004 | 2021 | $0.1 \pm 0.5$ | $0.3 \pm 0.5$ |
| Balearic Islands | 2004 | 2021 | $-4.5 \pm 2.1$ | $-1.5 \pm 1.1$ |
| Northen Sector | 2003 | 2021 | $-4.8 \pm 5.4$ | $-4 \pm 5$ |

*3.5. Sea Level Trends*

Ref. [8] analyzed the sea level at the tide-gauges of Ceuta, Tarifa, and Algeciras in the Strait of Gibraltar and Málaga, Alicante, and L'Estartit in the Mediterranean Sea. Gaps in these time series were filled by means of linear regression on nearby tide-gauges. In some cases, redundant tide-gauges, operated by different institutions, were used for filling such gaps. In this way, sea level time series as long as possible were constructed, in some cases dating from the 1940s. Altimetry data were used for the most recent period 1993–2020. Table 6 shows the linear trends for the sea level time series from tide-gauges, using the longest available series and from altimetry data for the period 1993–2020. Linear trends from tide-gauges series for the reduced period from 1993–2020 are also included for comparison. The results from tide-gauges are corrected for the effect of the GIA (Glacial Isostatic Adjustment [54]).

**Table 6.** Linear trends from tide-gauge data for the period 1948–2019 (column 3) and for the period 1993–2019 (column 4). These trends are corrected for the effect of GIA. Column 2 shows the contribution of GIA to relative sea level. Column 5 corresponds to the trends calculated from altimetry data at the grid points closest to the tide-gauge locations. * Palma results correspond to the shorter period 1997–2019.

| Trend (mm/yr) |  | Tide-Gauge | Tide-Gauge | Altimetry |
|---|---|---|---|---|
| Location | GIA | 1948–2019 | 1993–2019 | 1993–2019 |
| Tarifa | $-0.2$ | $1.4 \pm 0.2$ | $4.7 \pm 0.7$ | $2.5 \pm 0.3$ |
| Algeciras | $-0.2$ | $1.0 \pm 0.1$ | $2.3 \pm 0.6$ | $2.4 \pm 0.4$ |
| Ceuta | $-0.2$ | $0.9 \pm 0.1$ | $1.9 \pm 0.6$ | $2.4 \pm 0.4$ |
| Málaga | $-0.2$ | $1.4 \pm 0.2$ | $3.7 \pm 0.7$ | $4.1 \pm 0.4$ |
| Alicante | $-0.2$ | $0.8 \pm 0.2$ | $2.0 \pm 0.8$ | $3.0 \pm 0.3$ |
| L'Estartit | $0.1$ |  | $2.7 \pm 0.8$ | $2.7 \pm 0.3$ |
| Palma | $0.3$ |  | $2.0 \pm 1.1$ * | $1.8 \pm 0.5$ * |

The tide-gauge data show a positive significant trend in all sites with values ranging from 0.82 mm/yr in Alicante to 1.40 mm/yr in Málaga for the period 1948–2019. Computing the trends for the last three decades (1993–2019) shows larger values from 1.9 mm/yr in Ceuta to 4.7 mm/yr in Tarifa. This acceleration is also supported by altimetry data (see last column in Table 6). Moreover, looking at the trends computed from the coastal reconstruction of [7] for the same period, it can be seen that this is a generalized behavior in the WMED (see Figure 12). The reconstruction shows values ranging from 2.01 mm/yr close to the Gibraltar Strait to a maximum value of 3.15 mm/yr around Valencia. * Palma time series extended since 1997.

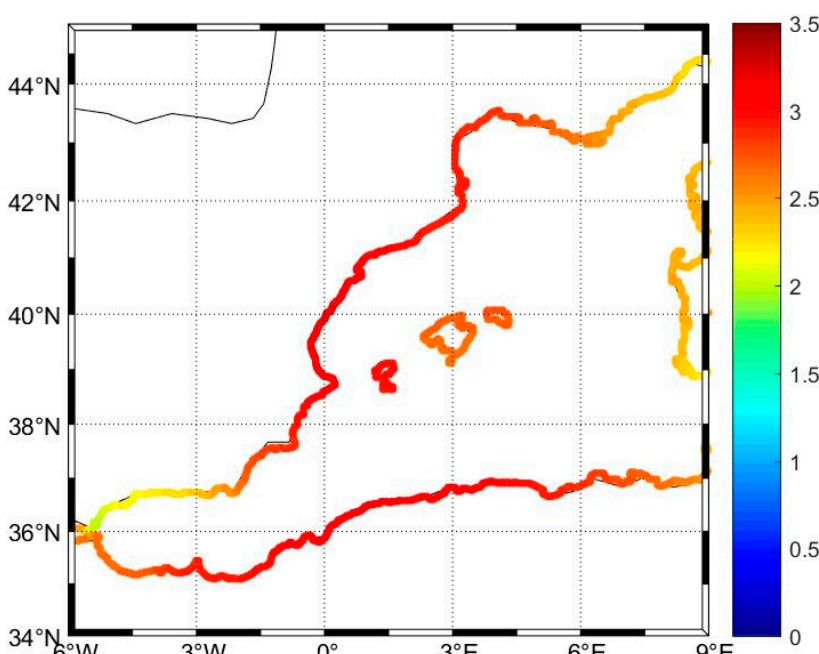

**Figure 12.** Coastal sea level trends (in mm/yr) estimated for the period 1993–2019 from the reconstruction based on tide-gauge data by [7].

Ref. [8] used a statistical approach and tried to decompose these trends into its different contributions: atmospheric forcing, including atmospheric pressure and meridional and zonal components of the wind, and thermosteric and halosteric contributions. The part of the trends not explained by these factors would be attributed to the increase in mass. This latter contribution could also be divided into the salt and freshwater contributions. Detrended monthly time series of sea level were linearly regressed on these contributing factors (atmospheric forcing, thermosteric, and halosteric contributions). The results showed that there was no significant correlation between sea level and the thermosteric or halosteric contributions in some of the locations analyzed. Sea level in Málaga was only correlated with the halosteric component, and sea level at Ceuta was significantly correlated only with the thermosteric component. The variability of sea level in Alicante was not correlated with the thermosteric, nor with the halosteric contribution. The two exceptions to this surprising and unreliable result was the time series of Tarifa and L'Estartit tide-gauge (see Table 4 in [8]). It is worth noting the high variance explained by the linear regression was in the case of L'Estartit tide-gauge where the multiple correlation coefficient reached a value of 0.88 (see Figure 13A). Black line in Figure 13A shows the sea level time series at L'Estartit tide-gauge. Red line is the sea level reconstructed by means of the linear model. Figure S1 in Supplementary Material shows the Alicante time series as an example where no correlation was found between the sea level and the steric contributions. The L'Estartit sea level time series is also included in Figure S1 for comparison.

The atmospheric forcing and the thermosteric and halosteric contributions explained 77% of the variance of the monthly time series of sea level at L'Estartit. The steric contribution was negative because of the large salinity increase of the water. The mass contribution was 3.3 mm/yr, this being divided into a contribution of mass of salt of 1.9 mm/yr and a contribution of fresh water of 1.4 mm/yr.

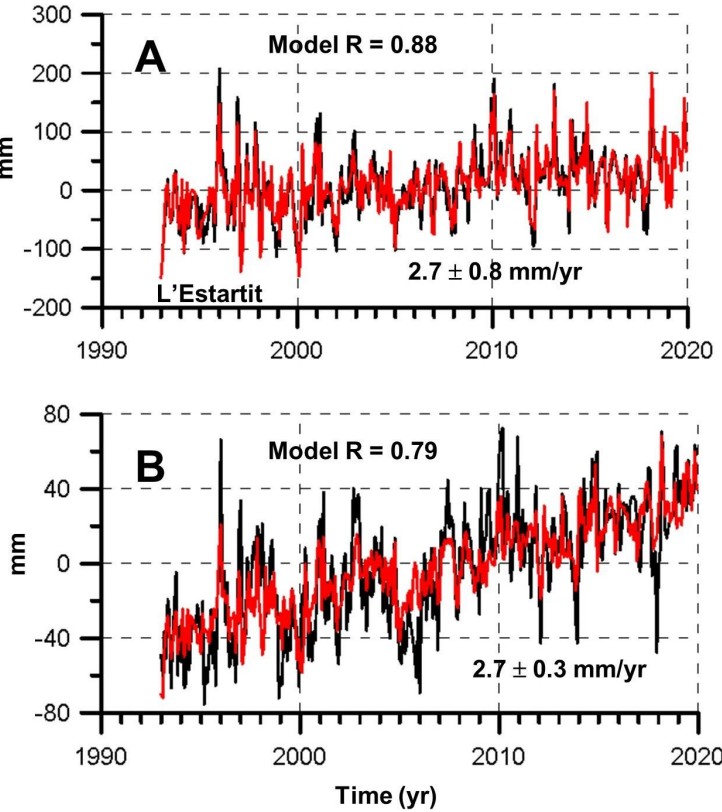

**Figure 13.** Black line in (**A**) is the time series of sea level at L'Estartit tide-auge. The red line is the reconstruction of sea level using a linear regression on atmospheric pressure, zonal and meridional components of the wind, and the thermosteric and halosteric contributions calculated from time series of temperature and salinity profiles. (**B**) shows similar results for the altimetry sea level in a grid point close to L'Estartit tide-gauge. Inserts in (**A**,**B**) show the multiple correlation coefficient and the sea level linear trend corrected for the effect of GIA.

*3.6. Reliability of Climatological Seasonal Cycles and Trends*

The analysis of long-term trends is based on the calculation of seasonal cycles that are subtracted from the original times series for obtaining time series of anomalies and the fit by least squares of a straight line. The three-monthly periodicity of the sampling, the frequent gaps, and the variability in the dates of the sampling could cast some doubts about the suitability of the sampling for the estimation of the seasonal cycles and the estimation of linear trends. These aspects are considered here (see Section 2.2.2).

A daily SST times series in a grid point close to Malaga Bay, in the Alboran Sea (4.5° W/36.7° N), was considered. The period covered was 1981–2021. A monthly time series was constructed averaging all the data within each month and year. Black dots and line in Figure 14A show the climatological seasonal cycle obtained from this time series. This cycle will be considered as a reference as it is derived from a daily series with no gaps. Vertical black lines are the 95% confidence intervals for the monthly mean values. In order to inspect the suitability of monthly samplings, we subsampled the daily series at the 15th of each month. Blue crosses in Figure 14A show the climatological seasonal cycle obtained from the monthly time series. It is clear that one data per month makes no difference for the calculation of the climatological seasonal cycle. Figure 14B shows again the seasonal cycle inferred from the daily time series as a reference. This time series was subsampled with a three-monthly periodicity. The sampling corresponded to the 15th of February, May, August, and November and had no gaps. Blue crosses show the four climatological seasonal values obtained from this subsampled time series. The four crosses lie on the curve defined by the reference seasonal cycle. The four climatological values of the three-monthly time series were interpolated for obtaining twelve climatological values

(red open circles in Figure 14B). The interpolated cycle did not deviate significantly from the reference cycle. Figure 14C,D show similar experiments. In the case of Figure 14C, we used three-monthly time series sub-sampled at different dates for each season of the year. The date when each data point was sub-sampled was chosen in a random way from the same distribution of dates as the real RADMED dates. This subsampling included gaps with the same frequency as the real RADMED dates. In Figure 14C, the three-monthly time series extended from 1981 to 2021. In the case of Figure 14D, the same procedure was applied, but the simulated three-monthly time series extended from 1993 to 2021 in order to simulate time series with the same length as the oldest ones from the RADMED project (see Section 2.2.2). The deviation of the four seasonal values and of the interpolated ones from the reference curve is larger for Figure 14C,D than for Figure 14B and the monthly sampling of Figure 14A. This result shows that the sampling step, the variability of the sampling dates, and the length of the time series affect the calculation of the climatological seasonal cycles.

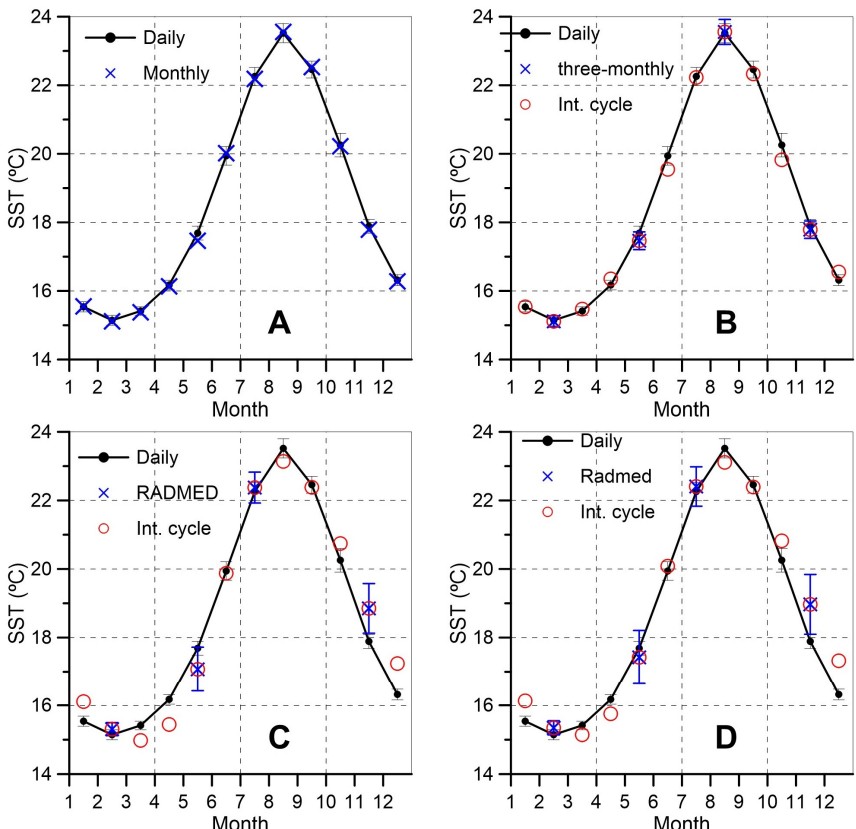

**Figure 14.** (**A**) shows the monthly seasonal cycles obtained from daily (black dots and line) and monthly (blue crosses) time series extending from 1981 to 2021. (**B**) shows the monthly seasonal cycle obtained from daily time series (black dots and line) and from three-monthly time series sampled at fixed dates from 1981 to 2021 (blue crosses). Red circles are the seasonal cycle made of 12-monthly values interpolated from the three-monthly seasonal one. (**C**) shows the monthly seasonal cycle obtained from daily time series (black dots and line) and from three-monthly time series sampled randomly at dates with the same distribution as dates from RADMED (including the possible existence of gaps) from 1981 to 2021 (blue crosses). Red circles show the interpolated seasonal cycle. (**D**) shows the monthly seasonal cycle obtained from daily time series (black dots and line) extending from 1981 to 2021, and from three-monthly time series sampled randomly at dates with the same distribution as dates from RADMED (including the possible existence of gaps) and extending from 1993 to 2021 (blue crosses). Red circles show the interpolated seasonal cycle.

Linear trends were calculated for the reference SST time series (daily) and for the subsampled ones. The trend for the reference series was 2.4 ± 0.5 °C/100 yr. In the case of the series subsampled with a monthly frequency, the trend was 2.4 ± 0.5 °C/100 yr. The trends for the series subsampled with three-monthly periodicity were 2.7 ± 1.1 °C/100 yr in the case of the series sampled at fixed dates with no gaps. It was 2.5 ± 1.8 °C/100 yr for the series sampled at random dates from the same date distribution as the RADMED project and extending from 1981 to 2021. Finally, the trend was 4.2 ± 4.0 °C/100 yr for the series subsampled three-monthly and extending from 1993 to 2021.

## 4. Discussion

The review of the works published since 2005 in the frame of the monitoring programs of the IEO offers a comprehensive (although sometimes incomplete) picture about the changes that the physical properties of the Spanish Mediterranean waters are experiencing as a consequence of climate change.

Temperature, salinity, and density time series from the RADMED project have been extended backwards in time using the MEDAR/MEDATLAS database. The results presented in this work update those of [3]. Previous analysis extended to 2015, and the present work has been completed until 2020 or 2021, depending on the series. We have reviewed the methodology used to construct annual time series for the estimation of long-term trends. This review shows that the data processing method can affect the final time series, and therefore, the estimation of trends. For this reason, the uncertainty associated with the choice of the data processing method should be considered, together with that linked to the natural variability of the oceanographic variables. Following this approach, we have found that the whole water column is increasing its salinity. Intermediate and deep waters are increasing their temperature, whereas no significant trends could be inferred for the temperature of the upper layer. This latter result may be a consequence of the scarcity of data, joined to the high natural variability of the surface temperature of the oceans. Ref. [55] subsampled simulated data from a numerical model at the same locations and times of real data in the Mediterranean Sea, and found that the available data, obtained without any temporal periodicity at scattered locations, were not suitable for the detection of long-term trends. In this work, we have presented a similar numerical experiment. In this case, we have used real daily data at a fixed position, simulating the sampling at one of the RADMED stations. This analysis shows that monthly sampling would be appropriate for both determining climatological seasonal cycles and linear trends for a variable with a high variance such as SST. The results obtained from a three-monthly sampling, at fixed dates and with no gaps, produced reasonably good results for the seasonal climatological cycle and yielded a linear trend, which was not statistically different from the real or reference one (that from the daily time series). Real sampling programs rarely can visit the oceanographic stations at the same dates of the year, or avoid the existence of gaps. These shortcomings decrease the quality of the seasonal cycles, although they still resemble the real ones, and increase considerably the uncertainty of the estimated trends. The trends from the three-monthly time series were not statistically different from the real ones, but the uncertainty of these estimations increased considerably.

Considering temperature time series with a higher time resolution, such as those from satellite measurements from the Copernicus Marine Service, from L'Esatartit station, or from the Fuengirola beach time series, the warming of the surface layer is clear and intense with trends higher than 2 °C/century. These time series extend over the periods 1971–2021, 1981–2021, and 1984–2021, respectively, and we cannot compare them directly with the joined time series of RADMED and MEDAR, which extend from 1945 to 2021. Therefore, the existence of warming trends in L'Estartit, satellite, and Fuengirola time series does not guarantee the warming of the Mediterranean surface waters during this larger period of time. However, L'Estartit time series has more than 50 years, and the analysis of subsampled time series discussed above makes us hypothesize that the lack of significant trends in the temperature of the surface layer arises from the scarcity of data. As the time

series gets longer, it is very likely that the warming of the Spanish Mediterranean upper layer will be statistically significant.

The present analysis has not revealed significant trends for the density of the upper and intermediate layers. In the case of the upper layer, once again we could speculate that this is a result of data scarcity. Nevertheless, this problem was not observed in the intermediate layer where the natural variability is much lower. It seems that the intense warming of the water has been compensated by an intense increase in the salinity. This would be in agreement with the lack of trends for the depth of the mixed layer in most of the stations and regions analyzed. This is an interesting result with important biological implications. An increase in the vertical stratification of the water column, as a consequence of the warming of the surface layer, would reduce the efficiency of the wind-induced mixing during autumn and winter months, resulting in a decrease in the primary production of the sea. According to present results, it could be considered that the salinity and temperature changes in the Spanish Mediterranean are compensating for each other with no effect, or very small, on density.

This salinity increase also affects the steric component of sea level, which has been found to be negative in the Spanish coastal waters. Nevertheless, the salinity increase makes a positive contribution, due to the addition of salt, to the mass contribution. These results are very clear in the case of L'Estartit sea level time series. Ref. [8] has shown that the monthly variability of sea level at L'Estartit can be reproduced by means of a linear regression on atmospheric pressure, zonal and meridional components of the wind, and the thermosteric and halosteric contributions calculated from monthly profiles of temperature and salinity (predictors or independent variables). These linear relationships were then used to estimate the contribution of each variable to the sea level rise. This analysis revealed that there was a mass contribution of 3.3 mm/yr that could be decomposed into 1.9 mm/yr of salt and 1.4 mm/yr of freshwater. It is surprising that this regression did not yield significant results for some of the other tide-gauges. This does not seem to be a reliable result, as the warming of the ocean has contributed significantly to sea level rise on a global scale [56]. We speculate that this lack of correlation between the two steric contributions and the sea level on a monthly time scale at some locations reflects once again what data scarcity already observed. This argument would be supported by the fact that the Northern Mediterranean Sea is a better sampling region, and therefore, the monthly grid product used to construct monthly time series of steric contributions would be more accurate.

In any case, the results presented along the last decades and reviewed in the present work highlight the importance of monitoring the water column for detecting changes associated with climate change. It is worth mentioning the fundamental role that autonomous profilers (with 5- or 10-day cycles) are playing by providing vertical profiles of temperature, salinity, and in many cases dissolved oxygen and fluorescence over the whole Mediterranean Sea from surface to 2000 m depth. However, this data source also has some limitations. Figure 15 shows the distribution of available Argo profiles in the four regions in which the Spanish Mediterranean has been divided in this work. It is worth noting that there are almost no data at sea depths shallower than 500 m, and no data at all in the continental shelf (200 m), with these areas of the sea being of paramount importance for the study of biochemical changes and other processes such as the contribution of steric component to coastal sea level.

This final result simply highlights that no single observing system is able to provide the complete information that is needed in order to fully understand the functioning of the oceans, their ecosystems, and the changes produced by climate change [57]. In situ sampling from oceanographic vessels is still needed to complement autonomous or remote-sensing measurements.

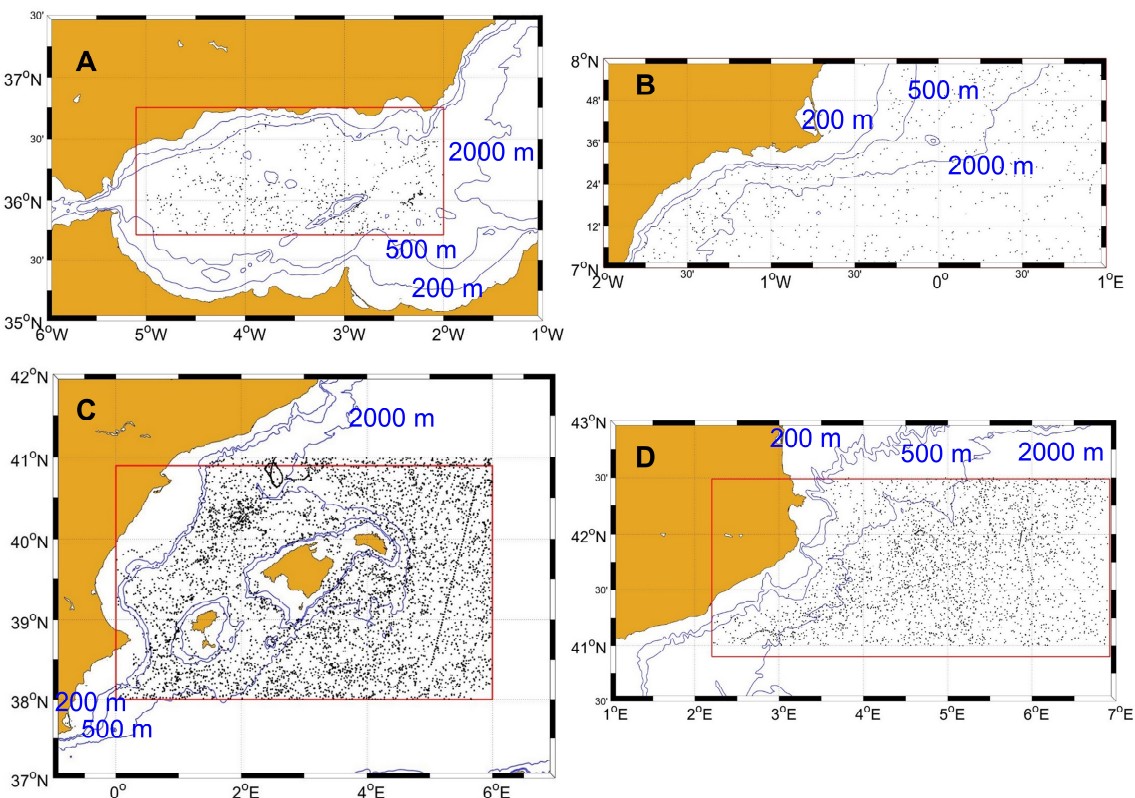

**Figure 15.** Blue dots correspond to all the temperature and salinity profiles available from Argo profilers in those areas analyzed in this work and represented by red squares. (**A**) corresponds to the Alboran Sea, (**B**) to Cape Palos, (**C**) to the Balearic Islands and (**D**) to the Northern Sector. The 200 and 500 m isobaths are included.

## 5. Conclusions

In summary, the Spanish Mediterranean waters exhibit increasing salinity along the whole water column at a rate ranging between 0.1 and 0.29100 yr$^{-1}$. The temperature of the intermediate and deep waters is also increasing. Using satellite SST or coastal data from L'Estartit oceanographic station and from the Fuengirola Beach time series, it could also be concluded that the surface layer is warming at a rate higher than 2 °C/100 yr. It is not clear whether or not the density is changing because salinity could compensate for the effect of temperature. In the case of the deep layer, the density is increasing.

The Spanish Mediterranean is absorbing heat since mid-twentieth century at a rate of between 0.20 and 0.72 W/m$^2$, a similar value to those reported for the global ocean [14].

No significant trends can be estimated in most of the regions analyzed for the MLD. This result would be related to the lack of significant trends for the density.

Sea level is increasing with an intensification of the trends since the beginning of the 1990s decade, with trends around 2.5 mm/yr.

The detection of the changes affecting the oceans require periodic sampling at fixed stations. The high variability of the sea, especially in coastal and continental shelf waters, obscures and makes it difficult to detect such changes in a statistically significant way. The larger the variance of the variable analyzed, the longer the length of the time series needed for the trend analysis. This work and previous ones have revealed that the scarcity of data, or the lack of a systematic sampling in the past, decreases the reliability of the results, making these problems more severe for short time series or in the presence of frequent gaps. These results highlight the importance of maintaining time series in a permanent and systematic way, reducing gaps as much as possible.

**Supplementary Materials:** The following supporting information can be downloaded at: https://www.mdpi.com/article/10.3390/jmse11071284/s1, Figure S1: Monthly time series of sea level (black line) and reconstruction by linear forward step regression on atmospheric pressure, zonal and meridional components of the wind, and thermosteric and halosteric contrinbutions to sea level (red lines). Upper plot corresponds to L'Estartit where the linear model chose all the potential predictors. Lower plot corresponds to Alicante, where the linear model only selected the atmospheric variables as potential predictors and the thermosteric and halosteric contributions were not statistically correlated to sea level. Table S1: Temperature, salinity, density, and heat content trends for the áreas: Alboran, Cape Palos, Balearic Islands, and Northern Sector.

**Author Contributions:** Conceptualization, M.V.-Y., F.M. and M.C.G.-M.; methodology, M.V.-Y., M.J. and G.J.; software, M.V.-Y., M.J. and G.J.; investigation, S.P., E.B. and C.A.; resources and data curation, S.P., R.S., R.B., M.S., J.S., J.P., E.T. and V.M.; writing—original draft preparation, M.V.-Y. and M.J.; writing—review and editing, F.M., E.B., J.S., E.T. and C.A.; visualization, M.V.-Y.; project administration, M.C.G.-M.; funding acquisition, M.C.G.-M. All authors have read and agreed to the published version of the manuscript.

**Funding:** RADMED project (Series Temporales de Datos Oceanográficos en el Mediterráneo) is funded by the Instituto Español de Oceanografía. Part of this work is supported by the EuroSea project, which has received funding from the European Union's Horizon 2020 research and innovation programme under grant agreement No. 862626.

**Institutional Review Board Statement:** Not applicable.

**Data Availability Statement:** The datasets presented in this study can be found in online repositories. The names of the repository/repositories and accession number(s) can be found below: http://www.ba.ieo.es/es/ibamar, accessed on 1 March 2022.

**Acknowledgments:** Marine heat waves of this study were computed using the following free and open access data from near real-time and reprocessed satellite products distributed by the Copernicus Marine Service (https://marine.copernicus.eu/, accessed on 1 March 2022).

**Conflicts of Interest:** The authors declare no conflict of interest.

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
