# Peer review of "Observations in the Spanish Mediterranean Waters: A Review and Update of Results of 30-Year Monitoring"

_jmse, doi:10.3390/jmse11071284_

Round 1

Reviewer 1 Report

Comments and Suggestions for Authors

The manuscript is relatively well written, well structured for the most part and it contains important and timely information that will be of interest to the audience of this Journal.

I have a major suggestion for revising the structure of the manuscript. I found section 2 long; because it attempts to include a long list of subsections describing the data AND then the methodology to process and analyze the data the entire section as presented loses strength.

I suggest Section 2. Data sources (or something similar as a title) and Section 3. Data processing and analysis.  Then Section 4 is Results, etc.

In addition, why include a subsection on data that are not used in the present study? That is subsection 2.6. If the idea is to 'mention' that these data sources exist, then just mention it in a short paragraph somewhere at the top of the section on data sources (a few sentences) but I suggest omitting lines 272-288,

It follows from these comments above that subsection 2.7 would be replaced with Section 3. Data Analysis, a section that would include what is now 2.7 and 2.8.  I believe this manuscript will be more clear this way.

Comments on the Quality of English Language

There is some need for editing this manuscript for clarity.  I have sent back the pdf file with suggestions and comments on it (please click 'comments' upon opening the document if necessary) that I hope the authors will find helpful.

In addition, I found that the use of 'could' and 'would' in the text from lines 335 (page 10) through 380 (page 11) is confusing.  Indeed many approaches 'could' be used to answer a question which 'would' lead to an answer but in the work presented eventually one approach 'is' used which 'result' is an answer.

Author Response

Thanks very much for your constructive review. Please, find below in bold the answer to your comments.

The manuscript is relatively well written, well structured for the most part and it contains important and timely information that will be of interest to the audience of this Journal.

I have a major suggestion for revising the structure of the manuscript. I found section 2 long; because it attempts to include a long list of subsections describing the data AND then the methodology to process and analyze the data the entire section as presented loses strength.

I suggest Section 2. Data sources (or something similar as a title) and Section 3. Data processing and analysis.  Then Section 4 is Results, etc.

Ok! We have restructure section 2. We have divided it in sub-section2.1 and sub-section 2.2. This latter has been named “data analysis methods” in a similar way to that suggested by the reviewer.

In addition, why include a subsection on data that are not used in the present study? That is subsection 2.6. If the idea is to 'mention' that these data sources exist, then just mention it in a short paragraph somewhere at the top of the section on data sources (a few sentences) but I suggest omitting lines 272-288,

We accept that the structure and objectives of the manuscript were not clear enough in the previous version. Following the reviewer’s suggestions, we have fully restructure section 2. We have made it clear in the last part of the new introduction that the goal of this work is to review the state of current observing systems from IEO and also from other institutions. This is the reason for keeping the description of Puertos del Estado observing system and the the observatory BBMO. In this way the reader has an idea about all the sampling networks in the Spanish Mediterranean.

It follows from these comments above that subsection 2.7 would be replaced with Section 3. Data Analysis, a section that would include what is now 2.7 and 2.8.  I believe this manuscript will be more clear this way.

Ok! We accept that this was not clear and we have completely restructure section 2. We have divided it into two main sub-sections:

2.1 Observation networks and data sets

2.2 Data analysis methods.

Within the section 2.1 we include all the observing systems beginning with those from IEO, following with those from other institutions that have been used in the present work. The those observing systems that have not been used, and finally other data sources.

Section 2.2 is divided in 2.2.1 for describing the construction of time series, and 2.2.2 devoted to the uncertainty sources in the estimation of trends and climatologies. We sincerely believe that now the structure and the goals of this work are clearer.

Comments on the Quality of English Language

There is some need for editing this manuscript for clarity.  I have sent back the pdf file with suggestions and comments on it (please click 'comments' upon opening the document if necessary) that I hope the authors will find helpful.

In addition, I found that the use of 'could' and 'would' in the text from lines 335 (page 10) through 380 (page 11) is confusing.  Indeed many approaches 'could' be used to answer a question which 'would' lead to an answer but in the work presented eventually one approach 'is' used which 'result' is an answer.

Ok! We have reviewed the English.

Reviewer 2 Report

Comments and Suggestions for Authors

Review of “Observations in the Spanish Mediterranean waters: A review and update of results from 30-year monitoring” (jmse-2369627)

Reported is a review of observations in the Spanish Mediterranean waters and updated analysis of the trend analyses associated with climate change. This is a useful contribution to the knowledge of ocean observations in this area. My main comment is on the lack of information for ocean circulation observations in this area. This can be added or discussed in the manuscript for completeness. My comments are aiming at improving the quality of the paper.  I would suggest the manuscript be accepted after some minor revision.

Major comment:

Reviewed in this paper are time series of water temperature, salinity, water level, etc, but not currents.  Circulation and flow information is important for oceanographic research and applications.  Are there any moored ocean current observations and HF radar current mapping in the Spanish Mediterranean waters? It would be good to add this part.  If not available, then a discussion of the need of such observations should be added to the paper. Note that in some U.S. coastal regions, moored velocity observations have been sustained for more than two decades, e.g., Weisberg et al. (2009), doi:10.1029/2009GL040028.

Minor comments:

The abstract can be more concise.

Line 47, “altercations” should be changed to “changes”.

Line 416, SST was already defined  (Line 55).

Line 791, it would be good to cite a relevant publication here: Liu, Y., H. Kerkering, and R.H., Weisberg (Editors) (2015), Coastal Ocean Observing Systems, 461 PP., Elsevier (Academic Press), London, UK.

Lines 819-820, I like this concluding remark!

Comments on the Quality of English Language

Overall good. There are a few some word choice issues that can be fixed.

Author Response

Thanks very much for your constructive reveiew. Please, find below in bold the answer to your comments.

Reviewed in this paper are time series of water temperature, salinity, water level, etc, but not currents.  Circulation and flow information is important for oceanographic research and applications.  Are there any moored ocean current observations and HF radar current mapping in the Spanish Mediterranean waters? It would be good to add this part.  If not available, then a discussion of the need of such observations should be added to the paper. Note that in some U.S. coastal regions, moored velocity observations have been sustained for more than two decades, e.g., Weisberg et al. (2009), doi:10.1029/2009GL040028.

 This is a very interesting point. In fact, SOCIB has coastal radars and it has been explained in the new section 2.1.5. Puertos del Estado also has some radars that have described in section 2.1.6. However, to our knowledge, there are no works dealing with the analysis of long time series of this type of data in the Spanish Mediterranean waters. Therefore, we simply mention its existence, to give an overview of all the monitoring capabilities in the Spanish waters, but we cannot present results related to these observations. We have tried to find a place in the text to mention the very interesting work by Weisberg et al. (2009), but as these type of data have not finally used in our work, we have not found the right place. Any suggestion would be welcome.

Minor comments:

The abstract can be more concise.

 We admit that we have not been able to shorten it.

Line 47, “altercations” should be changed to “changes”.

Ok! Done.

Line 416, SST was already defined  (Line 55).

Ok! We have reviewed this and other parts of the text where acronyms were used.

Line 791, it would be good to cite a relevant publication here: Liu, Y., H. Kerkering, and R.H., Weisberg (Editors) (2015), Coastal Ocean Observing Systems, 461 PP., Elsevier (Academic Press), London, UK.

Ok! We have included this references. Certainly, this book is absolutely in the line of our present work. Thanks for the recommendation.

Lines 819-820, I like this concluding remark!

Thanks!

Reviewer 3 Report

Comments and Suggestions for Authors

The review results are shown in the separate sheet.

Comments on the Quality of English Language

Author Response

We thank you very much for this very detailed and constructive reveiew. We think that we have followed most of your suggestions and we hope that the manuscript is much improved.

Please, find in the attached document the answer to all your comments.

Round 2

Reviewer 3 Report

Comments and Suggestions for Authors

see the attached file.

Comments on the Quality of English Language

Author Response

Once again we want to thank the reviewer for a thorough and very constructive review. We have followed most of the suggestions. There were some important mistakes that we think that we have corrected: The dates for the climatological cycles, and the interpolation of such cycles. This new version has also addressed some other questions such as a clearer rescription of the methodology used for the construction of the time series. Please, find attached a detailed answer to your comments.

Round 3

Reviewer 3 Report

Comments and Suggestions for Authors

see the attached file.

Comments on the Quality of English Language

Author Response

Once again we want to sincerely thank the reviewer for a very exhaustive and constructive review. We have followed most of the suggestions and we hope that the manuscript has improved. We also hope that we have fully understood all the points raised by the reviewer. In fact we have tried to do our best.

Please, find attached a detailed answer to all the questions.
